# Is There Still a Place for Threaded Spherical Acetabular Components in Modern Arthroplasty? Observations Based on an Average 14-Year Follow-Up

**DOI:** 10.3390/jcm14113683

**Published:** 2025-05-24

**Authors:** Marek Drobniewski, Bartosz Gonera, Łukasz Olewnik, Adam Borowski, Kacper Ruzik, George Triantafyllou, Andrzej Borowski

**Affiliations:** 1Clinic of Orthopaedic and Paediatric Orthopaedics, Medical University of Lodz, 90-419 Łódź, Poland; marek.drobniewski@umed.lodz.pl (M.D.); andrzej.borowski@umed.lodz.pl (A.B.); 2Department of Clinical Anatomy, Masovian Academy in Płock, 09-402 Płock, Poland; l.olewnik@mazowiecka.edu.pl (Ł.O.); k.ruzik@mazowiecka.edu.pl (K.R.); 3Department of Orthopaedics and Traumatology, Medical University of Warsaw, 02-091 Warsaw, Poland; adamborowskio15@gmail.com; 4Department of Anatomy, School of Medicine, National and Kapodistrian University of Athens, Goudi, 12462 Athens, Greece; georgerose406@gmail.com

**Keywords:** total hip arthroplasty (THA), screw-in acetabular cup, long-term outcomes, osseointegration, cementless fixation, developmental dysplasia of the hip (DDH)

## Abstract

**Background:** Total hip arthroplasty (THA) remains the standard treatment for advanced osteoarthritis, including for complex deformities. Innovations such as spherical screw-in acetabular components aim to enhance fixation and long-term outcomes. This study evaluated the long-term clinical and radiographic outcomes of cementless THA using such implants. **Methods:** A retrospective analysis was conducted on 277 patients (293 hips) who underwent THA with a screw-in acetabular cup (Aesculap/BBraun SC, Tuttlingen, Germany) and Antega femoral stem between 2005 and 2024. Patients were evaluated using the modified Merle d’Aubigné and Postel (MAP) score, Visual Analog Scale (VAS), and radiographic classifications, with implant survival assessed via Kaplan–Meier analysis. **Results:** The mean follow-up was 13.8 years. At the final follow-up, 58.7% of hips achieved excellent MAP scores, and mean VAS pain scores improved from 7.1 to 1.8 (*p* < 0.05). Implant positioning was within the Lewinnek safe zone in 77.1% of cases. Revision was required in 6.1% of hips, mostly due to aseptic loosening. The five- and ten-year survival rates were 98.3% and 94.0%, respectively. **Conclusions:** Spherical screw-in acetabular cups provide durable fixation and satisfactory long-term outcomes in THA, particularly for dysplastic hips, supporting their continued use with careful surgical techniques.

## 1. Introduction

Total hip arthroplasty (THA) has become a commonly performed procedure and remains one of the most effective interventions for managing advanced hip osteoarthritis. Advances in implant technology have broadened the range of indications, allowing for its successful application in more complex clinical scenarios. Today, THA is increasingly used not only in cases of secondary hip degeneration but also in younger, more active patients [1]. Severe cases of protrusion or post-traumatic, post-inflammatory, or dysplastic degenerative changes of the hip joint can be successfully treated surgically with joint arthroplasty. With the increasing life expectancy and higher functional demands of younger patients, THA has become increasingly common in a younger patient population [2]. This demographic shift is reflected in long-term projections: in the United States, the annual volume of primary THA is projected to increase by 174% by 2030 compared to 2005, highlighting the growing demand for durable and versatile implant solutions [3].

Advances in biomaterials and implant design have enabled effective hip arthroplasty even in cases of severe joint deformity. Innovations such as small-diameter acetabular components, narrow anatomical stems, modular systems, navigation, and robotics allow procedures in anatomically challenging conditions [4,5]. Current research continues to refine implant materials, coatings, geometry, and techniques to maximize long-term biofunctionality and implant survival [6].

Modern threaded acetabular cups have shown remarkable initial stability, a key factor in promoting successful osseointegration and achieving long-term secondary stability. These threads can feature sharp, serrated, or flat teeth, as well as various hybrid designs and sub-patterns, differing in parameters such as the number, size, surface area, curvature, thread pitch, and chip flute [7,8]. These design elements play a critical role in the cup’s insertion dynamics and its initial stability. For instance, sharp teeth provide more aggressive bone penetration, ensuring superior initial fixation, whereas flat teeth help distribute stress more evenly, reducing the risk of stress concentration. Additionally, the number, size, and curvature of the teeth influence how effectively the cup integrates with the surrounding bone [9].

The aim of this paper is to present the outcomes of total cementless hip arthroplasty procedures utilizing a spherical threaded acetabular component implanted using the screw-in technique.

## 2. Materials and Methods

### 2.1. Study Design and Inclusion Criteria

This retrospective study was conducted at our institution, where uncemented total hip arthroplasty (THA) has been routinely performed since 1985. Between 2005 and 2024, a total of 4170 THA procedures were carried out, of which 3582 (85.9%) utilized uncemented fixation. For the purposes of this analysis, only cases involving threaded acetabular components were considered, totaling 534 procedures. To be included, patients were required to have a minimum postoperative follow-up of eight years. Of the 341 patients who initially met this criterion, 38 were lost to follow-up. Ultimately, the study cohort comprised 277 patients who underwent 293 THA procedures, including several bilateral cases. All whose procedures had been performed using a screw cup (SC) spherical acetabular component by Aesculap/BBraun and an Antega anatomical femoral stem (Aesculap/BBraun, Tuttlingen, Germany) were eligible for analysis. Flowchart of the selection process is summarized in Figure 1.

### 2.2. Surgical Technique

All procedures were performed by four senior orthopedic surgeons (M.D, M.S, P.K, and T.D.) with at least 10 years of arthroplasty experience using standardized protocols. Surgeries were performed under epidural anesthesia via an anterolateral approach without greater trochanter osteotomy. Acetabular components were implanted within the Lewinnek-defined safe zone. Anteversion of the artificial acetabular component was maintained at ≤15° while the femoral stem was implanted with an anteversion of 5–10°. In most cases, the acetabular shell was supplemented with an asymmetric polyethylene liner. For patients under 50 years of age, using ceramic femoral heads with a diameter of 28 mm or 32 mm was standard practice.

Standard antibacterial and antithrombotic prophylaxis was administered perioperatively. Early mobilization began on postoperative day one with rehabilitation exercises. Upon removal of the Redon drain, patients initiated weight-bearing as tolerated. Progressive rehabilitation continued in subsequent days.

### 2.3. Implant Description

The SC acetabular component is a slightly flattened hemispherical cup featuring a central opening for implantation monitoring and potential graft supplementation. This opening is sealed with a titanium cover. The component’s inner surface is matte to prevent micromotion between the shell and liner. The self-tapping thread, extending throughout the cup’s height, is angled counter to its insertion and interrupted by vertical slots for depth control.

SC acetabular cups are available in 11 sizes (44–68 mm). The 44 mm and 46 mm cups require a 28 mm femoral head when paired with a polyethylene (UHMWPE, ISO-5834-2) or BIOLOX forte ceramic liner (Al_2_O_3_, ISO-6474). For sizes ≥ 48 mm, liners accommodate 28 mm or 32 mm heads. Polyethylene liners are available in symmetric and asymmetric (10°) versions, with the thinnest liner measuring 6 mm. The acetabular component is made from forged titanium alloy Ti6Al4V (ISOTAN F^®^, ISO-5832-3), featuring a rough, porous titanium coating (PLASMAPORE^®^) with a 0.35 mm thickness, increasing porosity by 40% (pore size: 50–200 µm).

### 2.4. Postop Assessment

Patients underwent clinical and radiographic evaluations preoperatively and during their final follow-up visit in 2024. Initial follow-ups occurred at 3, 6, and 12 months postoperatively, followed by annual visits.

Clinical outcomes were assessed using the Merle d’Aubigné and Postel classification, modified by Charnley (MAP) [10], evaluating pain, gait, and range of motion. Pain severity was measured with the Visual Analog Scale (VAS). To estimate implant longevity, Kaplan–Meier [11] survival analysis was used using STATISTICA 10.0 PL, with revision surgery as the endpoint. Confidence intervals were set at 95%, and survival rates at 5 and 10 years were reported. Radiographic assessment included the Kellgren–Lawrence classification [12] for preoperative evaluation and serial postoperative imaging. Anteroposterior and axial radiographs of the operated hip were obtained to assess implant positioning, osseointegration, heterotopic ossification, and migration (horizontal, vertical, and angular). Acetabular component integration was graded using the De Lee and Charnley classification [13] while femoral stem osseointegration was evaluated using the Gruen and Moreland classifications [14]. Additional assessments included axial positioning, vertical migration, cortical hypertrophy, and periosteal/endosteal ossifications in seven defined zones. All radiographic evaluations were performed by independent researchers (A.B and Ł.O) uninvolved in the surgical procedures.

The study was conducted in accordance with the Declaration of Helsinki, and approved by Ethics Committee of the Medical University of Lodz (NR RNN/195/24/KE, approval date: 10 September 2024).

### 2.5. Statistical Analysis

Statistical analysis was performed with IBM Statistics for MacOS IBM SPSS Statistics for MacOS, Version 29 (IBM Corp., Armonk, NY, USA). Nominal data between unpaired observations were compared using the Chi-square test while McNemar’s test was applied for paired observations. Normality was assessed with the Shapiro–Wilk test. Continuous variables were analyzed based on measurement type: unpaired measurements were evaluated with an independent *t*-test if normality was met; otherwise, the Mann–Whitney U test was used. A paired *t*-test was employed for paired measurements when normality was satisfied. A *p*-value less than 0.05 was considered statistically significant.

## 3. Results

A total of 196 women (70.8%) and 81 men (29.2%) were treated. In 152 cases (51.9%), the left hip joint was operated on, while in 141 cases (48.1%), the right hip joint was treated. The mean age of patients at the time of surgery was 59.08 ± 12.01 years (range: 26–87). The mean follow-up period was 5046.6 ± 887.9 days (13.8 years), ranging from 3157 days (8.6 years) to 7104 days (19.5 years). A detailed characterization of the study group concerning selected parameters, including the gender distribution, is presented in Table 1. A statistically significant difference arose at the mean age of surgery; specifically, male patients were significantly younger (Table 1).

The most commonly used stem size for females was 12 L, utilized in 33 cases, accounting for 16.02%. In males, the most frequent stem size was 13 L, used in 13 cases, representing 14.94%. The stem size used was statistically different between sexes (*p* = 0.011). The most commonly used femoral head size for females was S, applied in 98 cases, which corresponded to 47.57%. For males, the most frequent head size was M, used in 36 cases, accounting for 41.38%. No statistically significant association was found for femoral head (*p* = 0.690). The most frequently used acetabular cup size for females was 52, applied in 45 cases, representing 21.84%. In males, the most common cup size was 54, used in 17 cases, corresponding to 19.54%. The acetabular cup size was statistically significant different between sexes (*p* < 0.001).

Preoperatively, as expected, all patients demonstrated poor clinical and radiological outcomes. At a mean follow-up of 13.8 years postoperatively, the final results, assessed using the modified Merle d’Aubigné and Postel (MAP) classification, were as follows: an excellent outcome was observed in 172 cases (58.7%), good outcome in 71 cases (24.2%), and satisfactory outcome in 32 cases (10.9%). A poor outcome was recorded in 18 cases (6.1%). The mean improvement in the modified MAP scale was 6.9 points, which was statistically significant (*p* < 0.05). The MAP scale was not influenced by the gender (*p* = 0.410), or the side of the surgery (*p* = 0.195). However, a statistically important difference arose between the MAP scale and the ages of the patients (*p* < 0.001). Younger patients were associated with a higher prevalence of excellent/good outcomes while older patients were associated with satisfactory/poor results.

A total of 18 revision surgeries were performed in the study group. Of these, twelve cases involved the revision of the acetabular component only while four procedures included the replacement of the femoral stem only, femoral head, or polyethylene liner. In most cases, aseptic loosening was the primary reason for revision, likely resulting from the “undersizing” of the acetabular component or femoral stem. In four cases, revision surgery was limited to replacing the articulating components (femoral head and worn polyethylene liner). One case of the septic loosening of the entire prosthesis was recorded. This patient had originally undergone surgery for advanced osteoarthritic changes due to the developmental dysplasia of the hip (DDH). Following the diagnosis, she underwent a two-stage revision, with the initial removal of the prosthesis and implantation of an antibiotic-loaded spacer. After two months, upon the normalization of inflammatory markers, the final stage of revision was performed, replacing the spacer with a revision endoprosthesis. A good clinical outcome was reported at the five-year follow-up.

Additionally, six cases (2%) of femoral nerve palsy were observed, with symptoms resolving within two to seven months postoperatively. No cases of mortality, thromboembolic complications, or pulmonary embolism were recorded in the study group. A detailed breakdown of clinical outcomes according to the underlying pathology is presented in Table 2.

Apart from the aforementioned revision cases, no signs of the aseptic loosening of the prosthesis were detected on radiographic assessment. The proper placement of the acetabular component within the native acetabular region (True Acetabular Region, TAR), in the safe zone defined by Lewinnek [15], was confirmed in 226 cases (77.1%). In 12 cases (4.1%), mainly in patients with bilateral dysplastic osteoarthritis, significant bone loss in the acetabular dome necessitated the intentional superior placement of the acetabular component in the so-called secondary acetabulum. In sixteen cases (13%), implantation of the acetabular cup was outside the Lewinnek safe zone, with nine cases exhibiting excessive inclination and seven cases being too shallow. However, no radiological evidence of loosening was observed in these cases.

Additionally, sixteen cases (5.5%) of heterotopic ossification (HO) were noted, classified according to Brooker’s classification [16]: grade 1 in nine cases and grade 2 in seven cases.

Using the Visual Analog Scale (VAS) [17], preoperative pain levels averaged 7.1 points, while postoperatively, pain levels improved to 1.8 points. This pain reduction was statistically significant (*p* < 0.05).

Subjective patient-reported outcomes postoperatively were notably better than the objective results obtained using the modified MAP classification [10]. The greatest improvements were seen in pain relief and range of motion, contributing to enhanced hip function and overall patient satisfaction. As anticipated, the poorest functional outcomes were observed in patients with dysplastic osteoarthritis. However, it is important to note that an “excellent outcome” in the modified MAP classification represents a result comparable to a healthy native hip joint. Importantly, no cases of thigh pain—which is sometimes reported following uncemented hip arthroplasty—were recorded in this study.

Implant survival probability was assessed using the Kaplan–Meier estimator [11]. The five-year survival probability for the entire endoprosthesis was 97.6%, while for the SC acetabular component alone, it was 98.3%. For the ten-year follow-up period, implant survival was estimated for 235 hip joints. The ten-year survival probability of the entire prosthesis was 91.5% whereas the SC acetabular component alone had a survival probability of 94%. A detailed breakdown of implant survival probabilities is presented in Figure 2 and Table 3. 

## 4. Discussion

The design of screw-in acetabular components has undergone multiple modifications over the years. First-generation screw-in cups relied solely on mechanical stabilization within the bone bed, featuring a metallic, smooth outer shell without any additional coating or porous surface. While this design provided adequate primary mechanical stability, it failed to create optimal conditions for osseointegration, thus compromising long-term biological secondary stability [18]. This limitation is evident in the poor long-term outcomes observed in total hip arthroplasties (THAs) utilizing Parhofer–Mönch-type screw-in acetabular cups.

The second generation of screw cups introduced porous coatings and hydroxyapatite spray on the outer surface of the acetabular shell, significantly improving conditions for secondary biological stability and ensuring the long-term osseointegration of the implant within the acetabular bone bed [19].

The third generation of screw cups focused on enhancements to the liner, including metallic, ceramic, or polyethylene options, as well as improvements in polyethylene materials and sterilization techniques to enhance wear resistance [20].

Experimental laboratory studies on force distribution around different acetabular components have demonstrated the superior primary stability of screw-in acetabular cups reducing micromotion and promoting osseointegration, especially in the retroverted, protrusion, or shallow acetabulum, making them advantageous in select complex primary arthroplasty cases compared to press-fit cups implanted by impaction techniques [21].

The SC acetabular cup features thin load-bearing thread surfaces with anti-slip edges, reducing the force required to cut into the bone. Additionally, the large thread pitch enables the engagement of stronger bony trabeculae, enhancing initial stability. The counter-angled thread surfaces allow for the even distribution of insertion forces, preventing localized stress concentrations [22].

During implantation, once the base of the thread reaches full bone contact, the cutting grooves anchor firmly into the bone tissue, leading to a significant increase in torque resistance until proper implant depth is achieved. Unlike conical screw-in cups, the spherical design of the SC acetabular cup preserves the subchondral bone layer, which may contribute to better long-term fixation [23].

A review of the literature indicates that THA procedures utilizing screw-in acetabular components continue to attract significant research interest. The Zweymüller-type THA with a conical screw-in acetabular cup remains the fourth most commonly used implant worldwide. Several key studies support the clinical relevance of these implants. While there is extensive research on conical screw-in cups, recent publications on spherical screw-in acetabular components are relatively limited.

Christodoulou et al. [8] analyzed the early outcomes of 95 THAs using the titanium DELTA ST-C spherical screw-in cup. The mean patient age was 69.3 years, and the mean BMI was 27.4. The study group comprised 64% women. Using the Harris Hip Score (HHS), the study demonstrated an improvement from a preoperative mean of 46 points (range: 38–55) to a postoperative mean of 96 points (range: 94–100) at a 3-year follow-up, with no recorded revision surgeries.

Almeida et al. [24] examined outcomes of 301 THAs using the Tropic spherical screw-in acetabular cup in 268 patients with a mean age of 56.1 years (range: 27–75). At a mean follow-up of 16.9 years (range: 10.4–25), the final HHS was 83.3 while the modified MAP score averaged 15.3 points. The Kaplan–Meier survival rate was 81.2%. Due to a high polyethylene wear-related revision rate (56%), the authors shifted from screw-in to press-fit acetabular components.

Gala et al. [9] compared 150 third-generation screw-in cups to 150 press-fit cups in THA. The mean follow-up was 52.5 months (range: 27–78). No significant differences were noted in revision rates or clinical outcomes, assessed using the Hip Disability and Osteoarthritis Outcome Score (HOOS).

Ellenrieder et al. [25] compared 42 third-generation screw-in cups (Trident TC) with 42 press-fit cups (Trident PSL). The study groups showed no statistically significant differences in patient characteristics. At a 5-year follow-up, the HHS improvement was significantly higher in the screw-in cup group. However, scores based on WOMAC (Western Ontario and McMaster Universities Osteoarthritis Index) and SF-36 (Short-Form Health Survey) did not confirm this difference. Notably, the screw-in cup group showed more precise implant positioning, with a mean inclination angle of 48.4° ± 4.21° (range: 41–58°), compared to 54.4° ± 8.5° (range: 39–77°) in the press-fit group. Similarly, the anteversion angle was 17.3° ± 2.61° (range: 9–22°) in screw-in cups versus 22.1° ± 6.29° (range: 10–38°) in press-fit cups. No significant differences were found in complication rates.

In our study, the observed reduction in VAS scores from 7.1 to 1.8 represented not only a statistically significant change (*p* < 0.05) but also a clinically meaningful improvement in patient-reported pain outcomes. In the context of total hip arthroplasty, a decrease of two points on the VAS is generally regarded as the minimal clinically important difference [26]. Therefore, the magnitude of improvement observed in this study substantially exceeded this threshold, indicating a profound effect on patients’ pain perception and functional capacity. This finding underscores the therapeutic value of the implant beyond statistical metrics and supports its utility in improving quality of life following THA.

This study observed a revision rate of 6.1% at a mean follow-up of 13.8 years. When evaluated against current large-scale registry data, this rate aligns favorably with established benchmarks. According to a comprehensive international analysis of over 689,000 primary total hip replacements, the mean revision rate was found to be 1.29 revisions per 100 component years, translating to an expected cumulative revision rate of approximately 12.9% at ten years of follow-up [27]. Thus, the revision rate reported in our study remained well below this threshold, even with longer follow-up, indicating satisfactory implant longevity and clinical durability in the evaluated cohort.

Due to the heterogeneity of the examined cohort, there was a notably higher proportion of revisions observed in patients with the developmental dysplasia of the hip (DDH). This subgroup remains one of the most technically demanding in primary total hip arthroplasty. In our experience, the abnormal acetabular anatomy, poor bone stock, and frequent superolateral deficiency in the DDH increase the risk of inadequate initial fixation, especially when using cementless components. Although the screw-in cup offers an improved control of insertion depth and reliable fixation in shallow sockets, its performance is still dependent on bone quality and precise positioning. In cases with significant deformity, even slight undersizing or malorientation may predispose the construct to early loosening [28].

In 2021, our center conducted an analysis of 595 THAs using the Antega anatomical femoral stem [29]. The acetabular component was a press-fit Plasmacup in 307 cases and an SC screw-in cup in 288 cases. Both cups had the same polyethylene liner type. At a mean follow-up of 10.4 years (range: 5–15.5), there were 22 cases of the aseptic loosening of the acetabular component: 10 cases in the Plasmacup (press-fit) group vs 12 cases in the SC screw-in cup group. Further statistical analysis revealed no significant difference (*p* = 0.5714) in acetabular component loosening rates between the two groups.

Recently, in our center, we have also conducted an analysis of the only threaded acetabular component for which the manufacturers recommend impaction during implantation—the L-cup by Biomet. Based on 315 patients, with a mean age of 60 ± 11.25 years at the time of THA, who have had implanted L-Cups, we have observed prosthesis loosening in 20.5% of cases at the mean follow-up period 19.2 years (range 14–25 years). Our results have suggested that alongside proper patient qualification, preoperative planning, and surgical techniques, the selection of implants is also a key factor for successful THA [30].

Our findings align with the existing literature, reinforcing the clinical reliability of screw-in acetabular components. Based on our long-term experience, we believe that when precise surgical techniques are followed, screw-in acetabular cups can meet the expectations of orthopedic surgeons across various hip pathologies.

This study, however, had some limitations. Notably, it lacked a control group using alternative acetabular fixation methods, such as press-fit or cemented cups, which restricts direct comparisons and limits conclusions about relative effectiveness. Furthermore, due to the lack of documentation of some patient variables such as bone quality and activity levels, it was not possible to perform multivariate regression analysis to identify independent predictors of implant failure or poorer outcomes.

## 5. Conclusions

Spherical screw-in acetabular cups have demonstrated reliable fixation and long-term stability in total hip arthroplasty. Our findings support their continued use in selected cases, particularly in patients with challenging acetabular anatomy such as the DDH. Based on the results, we propose that the primary indications for spherical screw-in cups include the following:
-All acetabular components used in cases characterized by a deficiency of the bony acetabular ring such as in dysplastic coxarthrosis, particularly Crowe type II and III, where their fine-threaded design allows secure anchorage in the shallow dysplastic acetabulum without requiring full bone coverage.-The protrusion acetabulum, where the well-preserved acetabular ring provides adequate support for screw-in fixation, reducing the risk of acetabular floor fractures associated with press-fit implants.

Further long-term studies and comparative analyses are warranted to optimize implant selection and surgical techniques, ensuring the best possible outcomes for patients undergoing total hip arthroplasty.

## Figures and Tables

**Figure 1 jcm-14-03683-f001:**
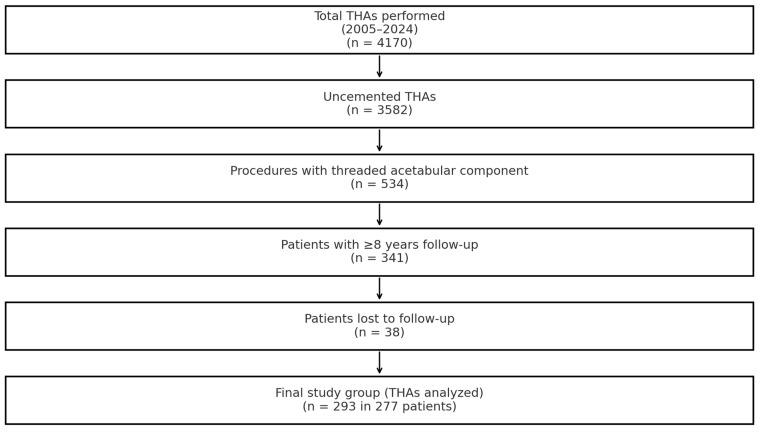
Flowchart of study cohort selection.

**Figure 2 jcm-14-03683-f002:**
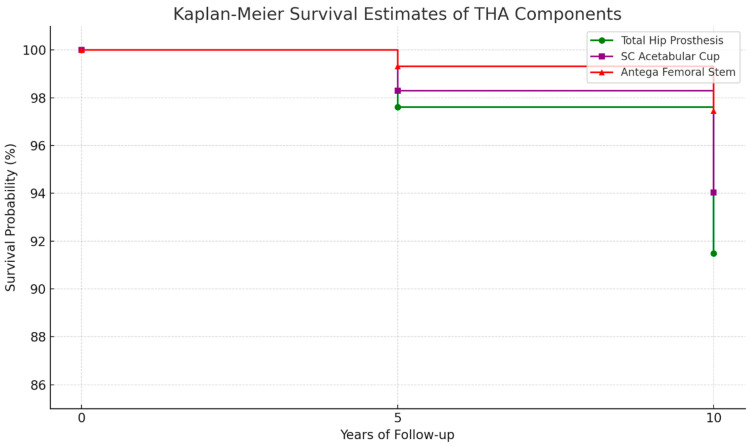
Survival analysis graph based on Kaplan–Meier analysis.

**Table 1 jcm-14-03683-t001:** Characteristics of the study group by gender distribution.

Parameter	Women (*n* = 196, 70.8%)	Men (*n* = 81, 29.2%)	*p*-Value	Total (*n* = 277)
Number of procedures	206 (70.3%)	87 (29.7%)	-	293
Left hip joint	103 (67.8%)	49 (32.2%)	0.322	152 (51.9%)
Right hip joint	103 (73.0%)	38 (27.0%)		141 (48.1%)
Bilateral (both left and right hip)	10	6	-	16
Mean age at surgery	60.17 ± 11.6 years	56.5 ± 12.7 years	0.047	59.1 ± 12 years
Min. age	32 years	26 years	-	26 years
Max. age	87 years	84 years	-	87 years
Mean follow-up period	5046.5 days (13.8 years)	5046.8 days (13.8 years)	0.883	5046.6 days (13.8 years)
Min. follow-up period	3157 days (8.7 years)	3224 days (8.8 years)	-	3157 days (8.7 years)
Max. follow-up period	7104 days (19.5 years)	7082 days (19.4 years)	-	7104 days (19.5 years)

**Table 2 jcm-14-03683-t002:** Final outcomes according to Merle d’Aubigné and Postel, modified by Charnley classification divided by etiology of the osteoarthritic changes.

Etiology	MAP–Excellent	MAP–Good	MAP–Fair	MAP–Poor
Idiopathic (188 hips)	141	35	11	1
Developmental Dysplasia of the Hip (DDH) (49 hips)	9	15	13	12
Protrusion acetabulum (23 hips)	9	3	3	2
Avascular Necrosis (AVN) (15 hips)	7	1	0	7
Post-traumatic (11 hips)	6	0	4	1
Inflammatory (6 hips)	0	3	1	2
Total	172	71	32	18

**Table 3 jcm-14-03683-t003:** Kaplan–Meier Biofunctionality Index for implants at 5 and 10 years of follow-up.

Implant Component	Kaplan–Meier at 5 Years (293 Hip Joints)	Kaplan–Meier at 10 Years (235 Hip Joints)
Total Hip Prosthesis	97.61% (99.38–95.84%)	91.49% (95.21–87.76%)
SC Acetabular Cup	98.29% (99.79–96.80%)	94.04% (97.16–90.92%)
Antega Femoral Stem	99.32% (100.26–98.37%)	97.45% (99.49–95.40%)

## Data Availability

The datasets used and analyzed during the current study are available from the corresponding author on reasonable request.

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
