# Peer review of "Is There Still a Place for Threaded Spherical Acetabular Components in Modern Arthroplasty? Observations Based on an Average 14-Year Follow-Up"

_jcm, 2025, doi:10.3390/jcm14113683_

Round 1
Reviewer 1 Report
Comments and Suggestions for Authors
discuss clinical significance but not just statistical outcomes (VAS improvements from 7.1 to 1.8)
Revision rate not critically evaluated; mention if this rate aligns with current acceptable benchmarks
there is too much general information on THA and biomaterials; condense what knowledge gap this study addresses regarding spherical cups
articulate the precise indication and advantages over other implants
The method section inadequately describes patient selection criteria and exclusion criteria clearly; outline patient selection and exclusion reasons
Implant survival at 10 years (94.0% for cup, 91.5% for whole prosthesis) presented without sufficient context; compare these outcomes to current literature and benchmarks for similar implants
High revision rate not deeply analyzed; discuss potential reasons - surgical technique or implant related factors contributing to revisions
Discussion lacks a detailed critique of the high revision and complication rates (especially in dysplastic cases
conclusions must be moderated to reflect study limitations realistically
authors recommend specific indications (dysplasia, protrusio) without sufficient evidence or robust comparative analysis
Reviewer 2 Report
Comments and Suggestions for Authors
Please note the correspondence author with “*” also in the list of authors.
The abstract is structured; please provide the country of the manufacturer Aesculap/BBraun SC (also in the methodology section); the keywords should be checked in accordance with MeSH.
In the introduction, discuss epidemiology and trend of THA in recent years, by referring to the scientific literature (for e.g. doi: 10.3390/medicina59020314 ). Objectives are stated at the end.
The methodology should be divided into subsections (for e.g. 2.1 Study design, 2.2 Data collection etc....). Provide a flowchart with the selection process. Add information on who performed the surgeries to exclude bias. Nothing is mentioned in the methodology about the Kaplan-Meier survival analysis in the methodology although results are presented with this analysis. Why did you not perfume a multivariate regression to identify independent predictors of revision or poor outcomes (for e.g. BMI, activity levels, or bone quality etc…)?
In the discussion section please define the limitations at the end.
In the results section please support Table 3 with a survival analysis graph. All tables should include an appendix section at the end. The conclusion advocates for screw-in cups in DDH, Table 2 shows DDH had the highest rate of poor outcomes (12/49 hips) – please explain. The high revision rate (6.1%) and aseptic loosening cases warrant deeper investigation; Were revisions linked to undersizing, surgical technique, or implant design? There are some questionable discrepancies in numbers (for e.g. e.g., 277 patients vs. 293 hips in Table 1). Line 141/142 – “presented in Table 1 (Tab. 1).” - it is not necessary to reference tables 2 times.
The conclusions support the main findings and propose future research directions.
The references are adequate but should be extended to reach a minimum of 30 given the type of paper.
Round 2
Reviewer 1 Report
Comments and Suggestions for Authors
The authors have made the requested changes. The work is ready for publication.
Reviewer 2 Report
Comments and Suggestions for Authors
The authors have improved their paper accordingly and respected all indications.